# Multiparameter optimisation of a magneto-optical trap using deep learning

A.D. Tranter[1], H.J. Slatyer[1], M.R. Hush[2], A.C. Leung[1], J.L. Everett[1], K.V. Paul[1], P. Vernaz-Gris [1], P.K. Lam[1], B.C. Buchler[1] & G.T. Campbell[1]

Machine learning based on artificial neural networks has emerged as an efficient means to develop empirical models of complex systems. Cold atomic ensembles have become commonplace in laboratories around the world, however, many-body interactions give rise to complex dynamics that preclude precise analytic optimisation of the cooling and trapping process. Here, we implement a deep artificial neural network to optimise the magneto-optic cooling and trapping of neutral atomic ensembles. The solution identified by machine learning is radically different to the smoothly varying adiabatic solutions currently used. Despite this, the solutions outperform best known solutions producing higher optical densities.

---

[1] Centre for Quantum Computation and Communication Technologies, Department of Quantum Science, Research School of Physics and Engineering, The Australian National University, Acton 2601, Australia. [2] School of Engineering and Information Technology, University of New South Wales, Canberra 2600, Australia. Correspondence and requests for materials should be addressed to B.C.B. (email: ben.buchler@anu.edu.au)

The interaction of light and atoms has long been a valuable test-bed for the foundations of quantum mechanics. The laser cooling of atoms[1,2] was a turning point that enabled a range of new and exciting developments in atom-light coupling techniques, shedding the complications inherent in the motion of free atoms. Cold atomic ensembles underpin many important advances such as the generation of Bose-Einstein condensates[3,4], cold atom based precision metrology[5], a new generation of optical atomic clocks[6] and quantum information processing[7–10].

In general, the efficacy of cold atomic ensembles is improved by increasing the number of atoms and reducing the temperature as this will increase signal-to-noise ratio of any measurement of the atoms. In the particular example of quantum information processing, the key metric is the resonant optical depth (OD) of the ensemble. The larger the OD, the stronger the atom-light coupling which is essential for maintaining coherence while mapping into and out of the atoms. The most prevalent cold atomic system is that of the magneto-optical trap (MOT) in which thermal atoms are collected into the trap from the surrounding warm vapour. OD is highly dependent on atomic species and trap geometry, however there exists a number of strategies to improve this characteristic, such as transient compression stages within an experimental run[11,12], polarisation gradient cooling[13] and temporal/spatial dark spots[12]. Despite the extensive amount of work done on laser-cooled atomic systems[14], it remains a challenging endeavour to construct a quantitative description that captures the complete atomic dynamics. This is mostly owing to the fact that these systems generally present computationally intractable dynamics in three-dimensions, involving many body interactions, polarisation gradients and complex scattering processes[15,16]. Furthermore, analytical models fail to account for experimental imperfections that may perturb the system. As such most strategies to improve optical depth are in general limited to intuition regarding adiabatic and monotonic approaches. However there have been indications that solutions outside of this space may lead to more efficient collection of atomic ensembles[17]. Recently it was also demonstrated that it is possible that BECs may be distilled from cold ensembles without evaporative cooling techniques by using a specialised compression sequence[18].

Human intuition is often used in laboratories to optimise experiments. Recent online experiments have also successfully used humans for tuning the parameters of some quantum simulations[19,20]. There are circumstances, however, where it is worth looking for solutions that are not in keeping with human expectations. For such systems, optimisation via an algorithmic process designed only to minimise a cost function can identify solutions that are highly non-intuitive and yet outperform traditional solutions. Machine learning techniques, in particular those based on deep neural networks ("deep learning"), have shown great promise for solving complex problems beyond human performance[21,22]. These techniques have been used to optimise the control of quantum experiments and theoretical protocols[23–28]. While these works demonstrate the efficacy of deep learning, they were performed offline, meaning that they acted on prior information rather than interacting with an experiment in real-time. Approaches to using machine learning for online optimisation of cold atomic systems have included Gaussian process models[29] and evolutionary algorithms[30–34], however, these techniques do not leverage the computational speed that deep learning has shown when working with large datasets.

In the current work we seek to optimise the OD of a cold atomic ensemble using a feedback-like online procedure. The algorithm is handed control of a set of experimental parameters that it tunes to optimise system performance. We use three independent artificial neural networks (ANNs) that together form a stochastic artificial neural network (SANN). Our approach is sufficiently computationally efficient to optimise a system with a large number of control parameters in real time.

## Results

**Machine learner design**. We use densely connected multilayer perceptrons as models of the MOT response to 63 independent piecewise experimental parameters (three experimental control variables divided into 21 time bins each). The network topology consists of a 5-hidden layer network with 64 neurons each, which can be trained in under one second on standard hardware (Intel i7-920 2.67 GHz). Our choice of activation function is the Gaussian error linear unit, which yields fast training and smooth landscapes[35]. The ANNs are trained using the Adam algorithm, which adaptively sets the step size. The early stopping technique is used to avoid over-fitting.

The initial training data is gathered using a differential evolution (DE) algorithm biased towards exploration for $2N$ points, where $N$ is the number of parameters. This means the atom compression sequence is run $63 \times 2 = 126$ times. Each of the three ANNs is then initialised with random neuron weights and finds a fit to the parameter landscape based on the training data. After this point, the machine learner runs in a loop of three steps. In step 1, each of the three ANNs is used to predict the values of the 63 experimental parameters that will minimise the cost function using the L-BFGS-B algorithm[36]. In step 2, each of the three parameter sets are applied to the experiment in turn, with an additional fourth parameter set obtained again using DE to ensure the ANNs continually receive unbiased data. In step 3, the outputs of these four experiments are used to update the neuron weights of the three ANNs. These steps are repeated until no further improvement in the cost function is seen (see Methods section for further details).

Using multiple ANNs is a way to ensure effective exploration of the parameter space. This kind of network aggregation[37] forms a SANN, where the stochastic element is introduced by the random initializtion of network weights and the extra step of DE. Having three ANNs facilitates exploration of the parameter space by ensuring the SANN does not become confined in local minima.

**Experimental apparatus**. We apply our implementation of a SANN to experimentally optimising the optical depth (OD) of a cold atomic ensemble. Our system comprises of a $^{87}$Rb elongated MOT as shown in Fig. 1 used for quantum memory experiments. The elongated shape provides the high OD required for such quantum memory experiments and is achieved via 2D elongated trapping coils[38,39] and additional capping coils for axial confinement. A detuned probe beam is sent through the atomic ensemble as a quick proxy for measuring OD (see Methods section). This measurement is fed back into the main optimisation loop so that the next set of parameters may be selected. We initially optimised the compression sequence following the conventional monotonic approaches used in other works[12,38,40] (See Fig. 2a). As this period of the experiment is crucial to the final optical density of the ensemble and spans a relatively short time compared to other periods within an experimental run, this sequence was chosen as the platform for online optimisation. We divide our compression sequence into 21 sequential time bins with a period of 1 ms each. We provide the SANN with arbitrary experimental control of the trapping field detuning, repump field detuning and magnetic field strength during each of these time bins giving a 63 parameter optimisation. Other than the bounds imposed by the physical limits of the experimental setup we

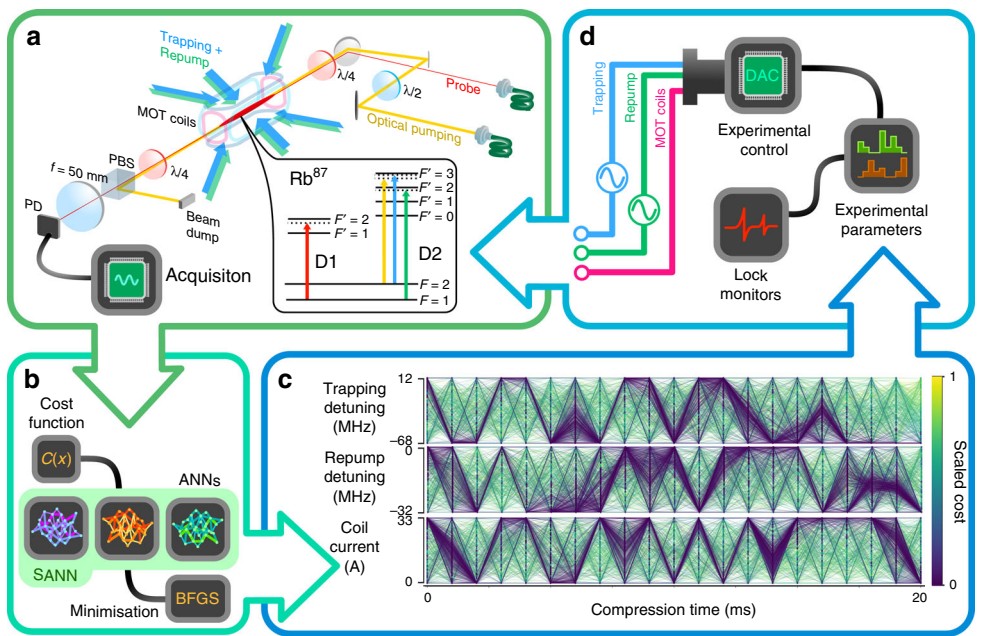

**Fig. 1** Online optimisation of optical depth. **a** Initially a MOT captures thermal $^{87}$Rb atoms via laser cooling. The ensemble is then transiently compressed using a set of 21 time bins for trapping frequency, repump frequency and magnetic field strength. The off resonant OD is measured from the transmission of a probe field incident on a photo detector. This value is passed to the SANN (**b**) where a cost function is calculated for the current set of parameters. Each ANN that comprises the SANN is trained using this and the previous training data. Each ANN generates a parameter set by minimising the predicted cost landscape using the Broyden-Fletcher-Goldfarb-Shanno (BFGS) algorithm. An example of parameter sets that were tested during an experimental run is shown in **c**. Each line represents one set of parameters that was tried and is coloured by the corresponding measured cost. Each predicted parameter set is sequentially passed to (**d**), the experimental control systems which monitor the lock state of the experiment and convert the parameter set to physical values. This loop continues until either a minimisation condition or maximum run number is reached

impose no constraints on parameter values during the optimisation process. These parameters are expected to have the largest effect on the transient compression of the ensemble as each parameter contributes directly to physical characteristics of the ensemble and trap such as scattering rate, cooling rate, velocity capture range and density. Furthermore, the SANN also has control of the detuning during the polarisation gradient cooling stage by way of setting the final value of the trapping detuning at the end of the compression sequence. In general we are limited by the experimental duty cycle and not computational power with 63 parameters providing a good trade-off between ramp granularity and the parameter landscape size.

The figure of merit or cost function for this optimisation is monitored via a photodetector which measures the incident probe field after absorption by the atomic ensemble. Large focusing optics are employed to eliminate the effects of lensing which may distort the measured OD profile, as this could allow a "gaming" of the cost function. As this optimisation is concerned purely with OD we construct a simple cost function

$$C(\boldsymbol{X}) = \frac{1}{P} \int_{t_i}^{t_f} p(t) \, \mathrm{d}t, \qquad (1)$$

where $C(\boldsymbol{X})$ is the cost for a given set of parameters $\boldsymbol{X}$, $P$ is a scaling factor derived from a reference signal to correct fluctuations in laser power and $p(t)$ is the measured photodetector response which is integrated across the pulse window. In this way a larger (smaller) cost is attributed to higher (lower) transmission of the incident probe light and thus lower (higher) OD. While this is not a strict measurement of OD, this inferred quantity affords a quick approximation which allows the optimisation speed to remain close to that of the repetition rate of the experiment.

The ultimate goal of our system, to have both a high optical depth and low broadening of the atomic transitions for use as a quantum memory, informs our experimental cycle. In addition to a high transient optical depth, the trapping magnetic fields must be turned off during measurements and the repetition rate must be as high as possible. Each cycle of the experiment contains four stages: a loading stage to collect atoms into the trap, a compression stage to transiently increase the optical depth, a preparation stage, and a measurement stage. During the preparation stage the trapping magnetic fields are turned off to allow eddy currents to decay and the ensemble is optically pumped into the $F = 1$ hyperfine state. We evaluate the cost function during the measurement stage. At the beginning of the next cycle, the loading stage re-captures the atom cloud, which is now falling and expanding.

We choose a repetition rate of 2 Hz as a compromise between running the experiment frequently and allowing a sufficient loading time. During each cycle, some atoms may be lost during the compression stage and, as the MOT takes longer to fill than the 500 ms loading time in each cycle, an equilibrium optical depth is reached after approximately ten full cycles of the experiment. Because we evaluate the cost function only after this equilibrium has been reached, the SANN will favour solutions that reduce atom loss between cycles as well as maximising the transient compression within the cycle. When applying each new set of parameters to test, the ensemble is completely released from the trap and we wait until the trap refills to a steady-state to ensure the next measurement is not biased by the previous parameter set. This wait time limits the rate at which we can test parameter sets to roughly once per 10 s. The optical depth is deemed to have reached equilibrium once the variation between repetitions is less than 2%.

**Comparison of machine and human optimisation.** An initial 126 training runs were collected which formed the training set

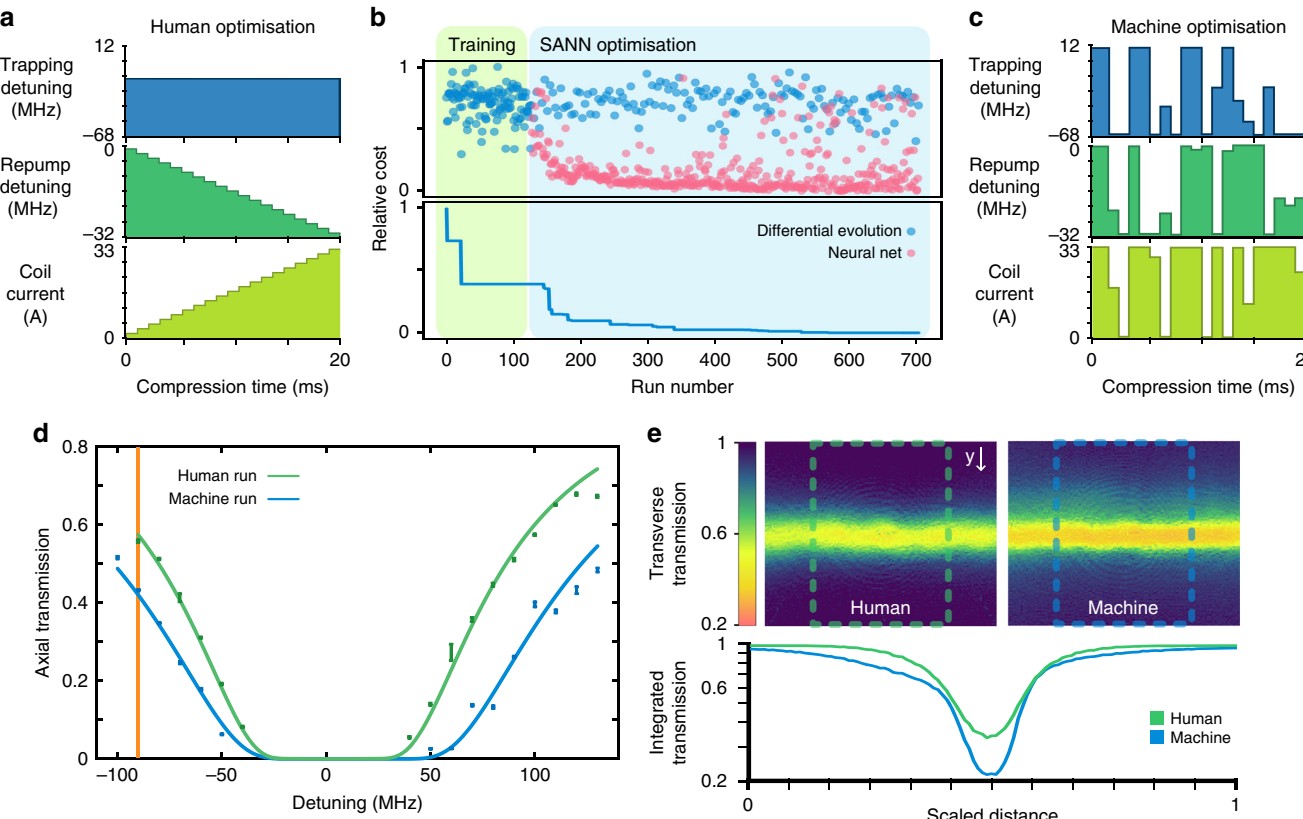

**Fig. 2** Experimental results using the SANN optimisation. **a** Human optimised compression stage using monotonic ramps for the magnetic fields and repump frequency (temporal dark SPOT). **b** Convergence of the SANN on an optimal solution after 126 training runs. The pink points are predictions generated by the ANNs while the blue points are generated by the differential evolution algorithm geared towards exploration. **c** Solution generated by the SANN for 63 discrete parameters that maximises off resonant OD by minimising the transmitted probe field. **d** OD measurements for the human and SANN optimised ensembles for a wide range of detunings corresponding to 530 ± 8 and 970 ± 20, respectively. The vertical line indicates the detuning used for optimisation. Errors are calculated from the standard deviation of 10 acquisitions acquired at each detuning. **e** Absorption images for the human and SANN optimised atomic ensembles. The lower log-plot shows a cross-section of the spatial distribution of the atoms which is directly influenced by each compression sequence. The cross-sections are integrated over the region indicated by the dashed boxes

used to concurrently train the ANNs. Following this a further 577 runs were recorded consisting of predictions made by the SANN. Convergence on the optimal solution occurred after approximately 580 runs as show in Fig. 2b, with the optimal solution given in Fig. 2c. It can be seen from Fig. 1c that our parameter space is not monotonic or flat, with a scaled cost of 1 corresponding to the absence of an atomic ensemble due to a failure to trap atoms with the parameter configuration.

We find that each of the ANNs converge to a distinct set of parameters corresponding to the optimal measured cost, while there exist many sub-optimal parameter sets which result in a less optically dense ensemble. Subsequent optimisation runs revealed that our parameter landscape contained many local minima which were within a few percent of the best presented solution. While different optimisations converged on these local minima we find that the relative effectiveness of each solution is constant irrespective of day to day variations in experimental conditions. From inspection of the human and SANN generated solutions shown in Fig. 2a, c, respectively, it is immediately obvious that the SANN solution is radically different to conventional techniques regarding temporal dark SPOTs[40] and transient compression schemes[12,38] that seek to increase atomic density. Instead we demonstrate a solution that exhibits structures that swing between the boundary values of our experimental setup and show little regard for continuity or monotonicity. While these piecewise ramps seem to defy physical intuition, we find that they

invariably outperform the best human solutions. The complex structures of the solutions hint at dynamics that are not well understood, although we speculate that they may be related to release-and-capture dynamics that have been observed in optical lattices[41]. The solutions generated by the SANN also indicate that extending the range of allowed values may yield improvements, however the parameters are scaled to the physical limitations of the experimental hardware in our current implementation.

As the measured cost is not a true measurement of OD we also sought to characterise the OD by mapping the absorption curve as a function of probe detuning. As shown in Fig. 2d we find good agreement between experimental data and the theoretical absorption given by:

$$I_t/I_0 = e^{\mathrm{OD}\frac{\gamma^2/4}{\Delta^2 + \gamma^2/4}}, \qquad (2)$$

where $I_t/I_0$ is the normalised transmitted intensity, OD is optical depth, $\gamma$ is the excited state decay rate and $\Delta$ is the probe detuning. We measure the OD of the best human optimised solution to be 535 ± 8 while the ANN optimised solution gives a measured OD of 970 ± 20, supporting the ANN solution as the optimal solution. The OD achieved by the human solution is on the order of our previously reported results used for high efficiency memory experiments[39] and represents the best achievable OD with our system using current methods. The ANN solution however affords an OD increase of (81 ± 3)% by

being agnostic to these techniques. Furthermore we find that absorption imaging through the side of the cloud shows a clear physical distinction between the solutions. We image using an expanded beam on the repump transition 9 MHz red detuned. As shown in Fig. 2e the ANN optimised solution has a higher density of atoms around the centre of the probe axis as well as increased homogeneity along the longitudinal axis. We also note that the atomic distribution is modified with a leading tail corresponding to a halo of atoms collected around the top of the atomic cloud.

## Discussion

Our work shows that a SANN is able to build a black-box empirical model of the cooling and trapping of neutral atoms, a process that is often difficult to model precisely using theoretical methods.

Using ANNs allows us to significantly increase the number of experimentally controlled parameters in our system. This is because the computational time required for an individual ANN training step can scale linearly with number of performed trials[42]. In comparison, the computational time required to fit a Gaussian process scales with the cube of the number of trials[43]. With a large numbers of parameters, such as we have in our work, the computational time required for a Gaussian process would quickly become large enough to limit the number of trials that could be done in the course of the optimisation sequence. Previous works have used ANNs to accelerate the convergence of differential evolution algorithms by using them as "surrogates" to quickly approximate a cost function[44–46] in classical control problems, such as microwave engineering[47] and airfoil design[48], however, in these cases, the evolutionary algorithm is put in charge of picking the next points in the experiment and balancing the exploration vs. exploitation trade-off[49]. Our system, by contrast, uses the SANN to directly and automatically optimise the system. We have not yet benchmarked the algorithm against other approaches due to the long experimental run times although we intend to do so in future work.

We find that the solutions are robust to daily fluctuations within the experiment and retain their relative efficacy. In general we are limited by the experimental duty cycle, so it is feasible that the SANN can be applied to experiments with larger parameter landscapes provided that they also support a higher duty cycle. We believe this would be well suited to applications with high dimensional structures such as imaging.

## Methods

**The magneto-optic trap**. The geometry presented allows for an atomic ensemble 5 cm in length containing approximately $10^{10}$ atoms at a temperature of $\approx 100\,\mu K$. Loading and cooling of thermal atoms into the trap is achieved using Doppler cooling with a field 31 MHz red detuned from the D2 $F = 2 \rightarrow F' = 3$ cycling transition. A field on resonance with the D2 $F = 1 \rightarrow F' = 2$ transition provides repump. A compression phase is then applied over 20 ms with 1 ms updates, which conventionally comprises of detuning the repump field to induce a temporal dark spot while monotonically increasing the magnetic fields to compress the trapped atoms into a smaller volume. However for the ANN optimised solution we switch the values every 1 ms following the sequence description presented in Fig. 2c. Following either approach, polarisation gradient cooling is applied for a further 1 ms for sub-Doppler cooling by detuning the trapping fields and switching off the magnetic confinement coils. Previous quantum memory schemes use a particular Zeeman coherence[39], which we realise by pumping into the mf = + 1 magnetic sublevel using a field 40 MHz red detuned from the D2 $F = 2 \rightarrow F' = 3$ during the application of a bias field applied axially along the ensemble. After a 1 ms dead time, which allows for the dissipation of eddy currents induced in the surrounding magnetic materials, we probe the OD of the ensemble using an axially propagating field red detuned from the D1 $F = 1 \rightarrow F' = 2$ transition. For optimisation purposes a detuning of −90 MHz was used to limit the effects of noise and lensing by the atomic ensemble. Our cost function spans a range of 57% absorption between the maximum achieved absorption at that detuning and a failure to trap atoms.

Optimisation is implemented by a control type feedback loop. The learner has control over the experimental parameters by feeding values to a field programmable gate array (FPGA) which sets the detuning of trapping and repump

fields using acousto-optic modulators while the magnetic field strength is set via a voltage controlled current source. Bounds on the parameters are monitored internally by the learner as well as by the control systems which are implemented in both Python and LabVIEW which transfer data via TCP sockets. Feedback is obtained via an acquistion card (NI 5761) which is passed to the learner to train the ANN and generate new experimental parameters.

The duty cycle of the experiment is approximately 2 Hz (see Fig. 3a) however, acquisition cannot occur at this rate as the experiment takes multiple runs to reach equilibrium. A single optimisation would generally take approximately 2 h at which point further acquisition was limited by experimental drift.

**The machine learner**. The machine learning algorithm is built into the Machine-Learning Online Optimisation Package (M-LOOP)[29,50], with the neural networks implemented using Tensorflow[51]. For a given optimisation, in the first 126 experimental runs the candidate parameters are chosen by a differential evolution algorithm (using the DE/rand/2 mutation scheme with scaling factor uniformly sampled from [0.5,1] at each generation, binomial crossover with rate 0.7, and population size $15 \times 63 = 945$). From this point on, a parameter set is generated by each of the three neural networks, then by the differential evolution, and then each neural network again, and so on in cycles of length four. To generate a candidate parameter set from a neural network, the network is first trained based on the data seen so far (see below for details of the training procedure). Then, the L-BFGS-B algorithm is run on the modelled landscape starting from 63 random locations, and the predicted best parameter set is selected. Combined with the use of multiple networks, this procedure effects an implementation of the Thompson sampling algorithm[52]. Each of the three networks is initialised using He initialisation (an improved version of Xavier initialisation). Training is performed using mini-batch gradient descent with the Adam update ($\alpha = 0.001$, $\beta_1 = 0.9$, $\beta_2 = 0.999$, $\varepsilon = 10^{-8}$) and batches of size 16. Training proceeds in iterations, each consisting of 100 (for the initial fit based on the initial 126 data points) or 20 (for subsequent fits) epochs, where one epoch is a single loop over the full data set. At the start of each iteration, a threshold is calculated as 80% of the current loss. At the end of that iteration, if the new loss is below that threshold then another iteration is performed. Otherwise, training terminates. The networks use $L^2$ regularisation with a coefficient of $10^{-8}$. Before being passed to the networks, all data are normalised using z-scores (based on the initial training data determined during the first 126 experimental runs). All hyperparameters (network topology, epochs per training iteration, training threshold and regularisation coefficient) were determined by manually tuning the algorithm on a random simulated 10-dimensional quadratic landscape.

Figure 3b shows the results of the other four optimisations, performed over multiple days, along with the ratio of the optical depth relative to the best parameter set. We expected that the different solutions could be due to day-to-day variations in experimental conditions, for example due to variations in ambient temperature or atom output from the rubidium dispensers, however we found that the ratios of the optical depths between the different solutions remained constant over the span of four months, even as the best optical depth varied from day-to-day. This suggests that the solutions are distinct local minima, which we verified by manually tuning some parameters around the optimal solutions, invariably resulting in a reduction of the optical depth.

**Network behaviour and convergence**. The SANN is initially trained on data generated by running the experiment with the differential evolution algorithm. This initial training will produce a rough model of the experiment, however it is susceptible to over-fitting due to the small size of the initial data set. This over-fitting will result in further exploration of the parameter space as the SANN incorrectly predicts cost minima. The SANN is continuously trained as more data is acquired, and will begin to sample more densely in regions where the cost function is predicted to be small. Regularisation and early-stopping are used to ensure that the SANN converges to an accurate model of the system in the vicinity of the predicted best operating points. The differential evolution algorithm provides additional exploration of the space, however the sampling density remains low away from the cost minimum.

A set of cross-sections through the cost landscape predicted by one of the trained models around the best observed solution is shown in Fig. 4a, b. The figure shows the predicted cost as each parameter is varied while the others are held fixed. In the vicinity of the predicted operating point, the model has a smooth cost landscape which suggests that the model is not suffering from overfitting.

We can gain insight into the performance of the SANN by monitoring the error between the predicted cost for each new parameter set and the cost that is subsequently measured by running the experiment. This provides a rough running estimate of the out of sample error for the SANN. Figure 4c shows the predicted and measured costs, out-sample error, and the distance from the optimal solution in parameter space, as a function of the number of trials. Initially, the SANN can be seen to have a high prediction error which decreases rapidly, along with the distance from the best operating point, as more data is acquired near the predicted minimum. The SANN occasionally predicts points far from the location at which the optimal point is ultimately found, with corresponding increased differences in the difference between the predicted an measured cost. We speculate that this is a result of over-fitting in regions with little collected data, providing some

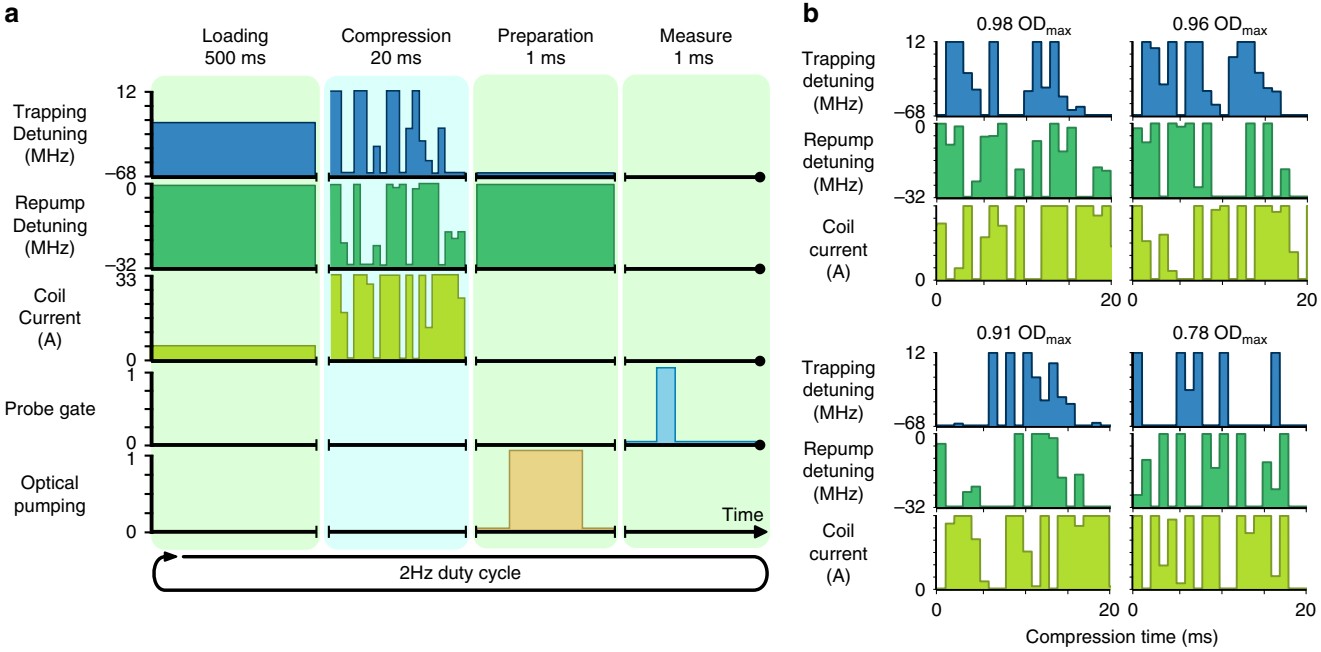

**Fig. 3** Experimental run and additional solutions. **a** A typical experimental run consists of four stages to generate an atomic cloud. Initially 500 ms worth of loading time is employed to statically load atoms into a trap. After this the compression sequence generated from a parameter set is performed over 20 ms. Following this a 1 ms preparation stage utilises PGC cooling and optical pumping schemes to the right magnetic level. Finally a measurement stage provides experimental feedback for the cost function via measurement of the probe absorption. The entire experimental run is repeated at a rate of approximately 2 Hz. The outcome of the measurement stage is monitored to determine when an equilibrium has been reached. **b** Additional sub-optimal parameter sets found by the SANN. The corresponding OD relative to the maximum OD achieved is shown above each solution

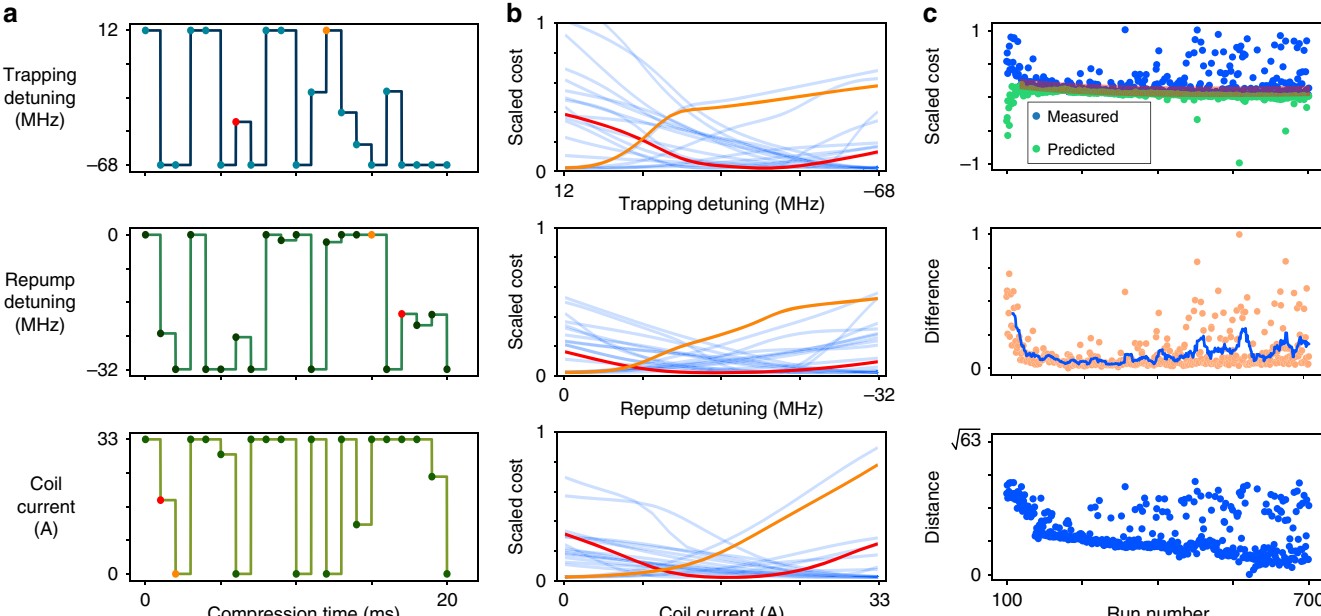

**Fig. 4** Cost landscapes and convergence predicted cost landscape cross sections generated by the model after exploration of the parameter space. Each 1-dimensional slice (**b**) is generated by varying each parameter independently over the available range while keeping other parameter values constant at their best known value shown in **a**. The red and orange curves represent arbitrarily chosen points that demonstrate landscapes for intermediate values and boundary limited values respectively. The blue curves represent every other parameter not highlighted in the experimental run. **c** The convergence of the model is attained by observing the measured and predicted costs as shown in the top plot. The red shaded area corresponds to experimental noise. The middle plot shows the scaled difference between these two measurements and the associated moving average as the SANN explores different regions of the cost landscape. The lower plot shows the distance of a given parameter set from the best observed parameter set

exploration of poorly sampled regions of the parameter space. Further exploration of the cost landscape is provided by the differential evolution algorithm, which contributes every fourth prediction (not shown on the plot).

**Code availability**. The machine-learning code used in this work is freely available[50] and is archived at https://doi.org/10.5281/zenodo.1435248.

## Data availability

The data that support the findings of this study are available from the corresponding author upon reasonable request.

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

## Acknowledgements

We thank D.B. Higginbottom for fruitful discussion. This research was funded by the Australian Research Council Centre of Excellence Grant No. CE170100012.

## Author contributions

A.T., H.S., M.H. and G.C. designed the study. A.T., J.E., A.L., K.P. and P.V-G. built and implemented the physical experiment and associated interfaces. H.S., M.H. and A.T. designed and built the model and optimisation code. A.T., G.C., M.H., B.B., P.K.L. and H.S. wrote the manuscript. All authors contributed to discussions regarding the results and analysis contained in the manuscript.

## Additional information

**Competing interests:** M.H. is currently employed by Q-Ctrl, a company developing control software for quantum technologies. The conception of this work and the experiments were conducted before M.H. started this role. All software discussed in this work is open source and in the public domain. The remaining authors declare no competing interests.

