## [Peer Review File · Nature Communications]

Reviewers' comments:

Reviewer #1 (Remarks to the Author):

Tranter and colleagues present the optimization of an atomic MOT compression stage using a stochastic artificial neural network. This approach is the first of its kind and builds upon previous work that optimized an evaporative ramp to BEC using a Gaussian process. The new approach, using an artificial neural network, allows the optimization of complex experimental protocols with many free parameters and is well-suited to ultracold atom experiments with short duty cycles. The automated optimization produces ultracold samples of atoms with an optical depth that is an impressive $\sim 1.8x$ greater than the human optimized case. Crucially, the experimental procedure for this improvement is vastly different to the typical compressed-MOT sequence. This empirical result provides new insight into the complexities of atom-light interactions and realizes a large, dense sample of ultracold atoms for quantum memory experiments. This improvement demonstrates the power of using neural networks to empirically optimize ultracold atom experiments and will be of interest to a general science audience as well as the ultracold community.

I believe that this publication should be published in Nature Communications provided that the authors address the following comments:

1. It was not immediately clear that Fig. 1 (c) presents actual experimental data. This only became clear after a comparison with Fig. 2 (c). The fact that this is the real cost landscape should be stressed in the text and/or figure caption.

2. It is clear from the text that a given compression sequence was repeated at 2 Hz and that atoms lost during this sequence were problematic as they would reduce the OD measured on subsequent repeats. This leads to several questions:

i. Assuming negligible loss during the compression - how many averages were performed for each compression sequence before moving on to a new parameter set?

ii. How much atom loss for repeats of a given compression sequence (at 2 Hz) was deemed acceptable? Were particularly poor choices of parameters sets (with a lot of loss) repeated with newly loaded MOTs at ~ 0.1 Hz?

iii. How do the statistical errors in the cost values (from these repeats of a given parameter set) compare to the markers used in Fig. 2b? If these error bars are significant they should feature in this plot.

A discussion of these points should feature in this paper, perhaps in the Methods section.

3. The authors should highlight if the 126 training runs were all for different parameter sets (at ~ 0.1 Hz) or also contained repeats (at 2 Hz) of specific parameters sets. At the moment this point is unclear to the reader.

4. On page 5, lines 3-4: "...we find that the relative effectiveness of each..." should be changed to read "...we find that the relative effectiveness of each...".

5. On page 5, line 5: "From inspection of the human and SANN generated solutions shown in Fig. 2a,c,..." should be changed to read "From inspection of the human and SANN generated solutions shown in Figs. 2a and c respectively,...".

6. It is clear from Fig. 2c that the current optimum sequence swings between the boundary values. The paper would be improved if the authors could pass comment on whether increasing the span

of these boundary values would potentially yield clouds with a higher OD.

7. Figure 2 would be improved if the y axis labels in Figs. 2d and e were given different labels rather than simply 'transmission'. This would stress that the transmission values in these plots were measured using different detectors, transitions and detunings.

Reviewer #2 (Remarks to the Author):

In this manuscript, Tranter et al. utilize neural networks to increase the optical depth of a cold atomic ensemble (in a MOT) by optimizing the compression sequence. The technique and results are interesting, and could constitute an important avenue for optimization of complex experiments. However, there are aspects of the experiment and result that need additional clarification (see below) before I can recommend the manuscript for publication in Nature Communications.

Note: In my comments below, I write from the perspective of an atomic physicist – I assume another referee with expertise in neural networks will comment more fully on the technical aspects of the neural network implementation used here.

****Major concern:** The major concern I have is understanding/justifying the irregular compression sequence found by the neural network. The authors state (or at least, strongly imply) that this sequence is the result of accounting for complex dynamics in the MOT. However, it is not clear to me that this is not simply a result of overfitting. Given that the network is dense and deep, there should be a very large number of variables/weights, which could overfit to the limited amount of training data (effectively allowing the network to individually account for each training data set, rather than extracting general optimization trends/processes). It seems that doing so would result in exactly the kind of irregular compression sequence that is observed, as well as many other local minima with nearly the same effectiveness (also observed). The observation that the optimal values at many of the sequence times lie on the boundary of the allowed values could also point to overfitting a limited data set. The fact that the early stopping technique had to be used is concerning for the same reason. Of course, there is no doubt that the network produced an improved solution – but whether that solution is accounting for complex atomic physics dynamics or just experimental variations in the training data (or other experimental aspects/imperfections difficult to measure directly) is not at all clear.

Additional questions/comments/clarifications (some related to the above):

(1) Boundary effects: The fact that the optimized values for the compression sequence are often at the boundary of each control variable's range is a concern. Have the authors considered the possibility that the true optimal solution actually lies outside of this range of values?

(2) Local minima: The authors note that there are many local minima with OD within a few percent of the "optimal" one. How similar are these other compression sequences to "optimal" solution presented? In the Methods, the authors note these had "slightly different parameter profiles" – please be more quantitative here. Presenting and discussing some of these solutions (perhaps in supplementary information) could be very useful.

(3) Human optimization: On page 2, column 2, 13 lines from the bottom, the authors note the "human optimized" solutions are linear ramps with the end points empirically optimized. Were other smoothly varying ramps (e.g. tanh, exp, gaussian) attempted? (Limiting the "human optimized" parameter profiles to linear ramps seems to limit the extent to which all adiabatic ramps can be excluded as non-optimal.)

(4) Parameters: The authors refer to “63 parameters” throughout most of the manuscript. For the reader, please clarify early in the manuscript (e.g., in the first paragraph on page 2) that these are 3 control variables, with values set at 21 distinct times (which together then constitute 63 overall parameters to optimize).

(5) pg. 2, column 1, end of second paragraph: It is unclear to me why the training time for the ANN scales linearly – please cite a reference here.

(6) pg. 2, column 2, 3 lines from bottom: The authors state here that the SANN has arbitrary experimental control, when it seems the values are actually limited to a certain range. Please clarify the statement here.

(7) pg. 4, column 1, line 8: Here the authors say the experiment repeats the compression sequence. Presumably, this is just to amplify the effect of the parameter set on the final OD, since I assume only one OD measurement is completed in each case (following the remainder of the experimental stages, such as polarization gradient cooling, etc.). Please clarify this here.

Reviewer #3 (Remarks to the Author):

The current work proposes a quantum control procedure to optimize the magneto-optic cooling and trapping of neutral atomic ensembles. The algorithm works based on the following steps (at least my understanding):

- 1- A set of control (learning) parameters are sent to the experimental set-up
- 2- An target (measured) value is assigned to the control (learning parameter)
- 3- A set of control parameters and corresponding target values are used to train three ANNs
- 4- A local greedy optimization algorithm (e.e. BFGS) is used to extract a more optimal parameter set from the three trained ANNs
- 5- The new parameter sets are sent to the experimental set-up to iterate the control loop till the a proper optimal set of control parameters are found.

The authors claim that the current work is able to find an optimal set of control parameters while an experimental run based on human intuition perform worse than the current approach.

General comments:

The current manuscript contains very interesting and valuable results that can vastly fuel the research within the field of quantum control and quantum information processing. I see the current approach wide enough to be used in any quantum control procedure. While using Deep Learning is shown to be valuable in this work, there is not enough evidence in the current work that this is the best way of solving such quantum control problems.

Of course using Deep Learning as part of the solution is novel, I think further steps have to be taken to elevate the quality of the current work to the level of nature communication journals. I believe the current manuscript can be considered for publication in nature communication journals if the following technical comments are addressed carefully.

Technical comment:

1- I did not find the flow of the manuscript in describing the steps of the proposed method that satisfying. For example authors explain how they use the Thompson sampling to choose the best model out of three ANNs that they train. However they fail to explain how they use it when it comes to the technical part of the algorithm. It also does not show-up in the Fig.1 as an important part of the control loop. This happened with the role of DE in whole procedure as well.

2- From what I understand authors use the whole dataset as training set without having a development or test sets. If this is true then I suspect that the convergence of the algorithms

come from ANNs because without having the test/dev sets then each ANNs will overfit to the training data and there is no guarantee that each would represent a fair approximate model of the cost function. On the other hand if authors have considered the train/test/dev set in their Machine Learning procedure then it has to be clearly explained in the manuscript.

3- In order to make the work reproducible one needs to know the set of hyperparameter used in this work. This includes the parameter for DE (mutation, crossover and so on) or the percentage of train/test/dev sets, learning rates and schedule if there is any and etc.

4- Fair comparison. One way authors can benchmark their method is to compare it with other quantum control procedures like, Krotov, GRAPE, DE , etc. Comparing the quality and speed of these algorithms with the proposed DeepLearning approach will gives us a better measure to judge the performance of the proposed method.

We would like to thank the referees for their careful reading of our submission. All three referees make interesting points about the manuscript, which we have revised accordingly to improve clarity and expand on the behaviour of the machine learning algorithm. Before addressing each comment individually, we will discuss the primary concerns that have been raised in common.

A point of confusion found by the first and second referees relates to the timing of the compression sequence and the repetition period of the experiment. The experimental sequence is informed by the requirements for using the MOT as a quantum memory. The compression sequence has been machine-learned within the constraints imposed by that goal. The MOT is compressed once per cycle, followed by a measurement of the optical depth, at a rate of 2 Hz. We repeat this cycle until the optical depth is stable from one cycle to the next, record the corresponding cost function, completely release the atoms from the trap, and repeat with the next set of parameters. To clarify this in the manuscript, we have added more detail about the timing sequence in both the main text and the methods section, including a timing diagram. We have also included an additional paragraph in the main text to provide context for what constraints informed our choice for timing in the loading-compression-measurement cycle.

Two other significant points that were raised are the related questions of what role overfitting plays in the learning process and how we validate the learned model. Because the deep neural network is very expressive and we have a small number of data points, particularly just after the initial training set of points has been collected, the model is very susceptible to overfitting. Initially, overfitting will lead to further exploration of the parameter space as the model makes incorrect predictions about the minimum of the cost function. At this point, the parameter space is undersampled and each of the neural networks in the SANN model overfit in different ways due to the random initialisations. This combination of overfitting with random initialisations implements a Thompson sampling method to efficiently minimise the cost function. As the parameter space is explored and more data is added to the training set the model will begin converging on the best parameter regions and will begin sampling more densely in promising areas.

Given that the model will initially overfit the data, it is interesting to consider whether the final trained neural networks accurately model the experiment. Before discussing this point, however, it is important to note that the goal of the optimisation procedure is not to produce a model of the experiment, but rather to find an operating point that yields an improvement. That goal was thoroughly achieved by the SANN learner. After machine optimisation, we used the best parameter set to run quantum memory experiments for four months with higher optical depths than anything we had achieved previously. The machine learner is now our go-to tool for solving the previously challenging task of optimizing optical depth after changes to the experiment. A task that used to take weeks of labour can now be accomplished in a matter of hours (during which the labourers are free to go for a coffee break).

Independently of the success of the algorithm, it is of course important to consider how well it models the experiment and whether the learned solutions reflect complex dynamics, or are artefacts of the model. We have included a figure in the methods section to elucidate how the model performs. Figure 4 (b) shows cross-sections through the cost landscape of the model corresponding to our best found solution. These show that the model is smooth in the vicinity of the optimal parameter set, indicating that the solutions are locally well-behaved, and does not show signs of overfitting. We expect that in areas of the parameter space corresponding to poor cost, however, the model would not accurately model the experiment because the space is sparsely sampled away from predicted minima. With this in mind, we didn't run test sets of parameters on the trained model because we neither expect nor require the model to produce accurate predictions in all regions of the parameter space. We can, however, compare the predictions made by the SANN with the corresponding measured cost functions for each trial of the experiment. Figure 4 (c) shows the predicted minimum cost and subsequent measured cost for each trial parameter set during an optimisation run. Initially, the algorithm performs poorly, predicting costs well below what the experiment produces. This is clearly a sign of overfitting. As the learner converges on a solution, however, the predictions improve. Interestingly, the SANN occasionally predicts minima that are well away from the best solution and perform poorly. We speculate that this is due to overfitting in areas of the parameter space that remain sparsely sampled, and result in exploration of the space away from the local minimum.

Below we detail the changes made to the manuscript in response to each of the referee comments. The

referee comments are in italics, followed by our responses.

Response to Referee 1

1. *It was not immediately clear that Fig. 1 (c) presents actual experimental data. This only became clear after a comparison with Fig. 2 (c). The fact that this is the real cost landscape should be stressed in the text and/or figure caption.*

The caption for Figure 1 has been modified to include a reference to the fact that the plot in Fig. 1(c) is experimental data.

2. *It is clear from the text that a given compression sequence was repeated at 2 Hz and that atoms lost during this sequence were problematic as they would reduce the OD measured on subsequent repeats. This leads to several questions: ...*

We have made modifications to the manuscript to improve our description of the experiment timing in the following ways:

- An additional paragraph has been added on page 4 to clarify the experimental sequence and provide context for why we have chosen the durations of the loading, compression, preparation, and measurement stages of the experiment.
- Additional text has been added to last paragraph on pg 4 detailing the repetition rate and the equilibrium condition.
- An additional comment has also been added regarding how an equilibrium condition is determined.
- Figure 3 has been added to the methods section which shows the full experimental run and details what is run at the fully duty cycle.

3. *The authors should highlight if the 126 training runs were all for different parameter sets (at 0.1 Hz) or also contained repeats (at 2 Hz) of specific parameter sets. At the moment this point is unclear to the reader.*

Additional text has been added to the methods section (3rd paragraph) detailing further hyperparameters in response to Referee 3. It also makes an additional statement that the first 126 runs for a given optimisation are chosen by a DE algorithm.

4. *On page 5, lines 3-4: "... we find that the relative effectiveness of each..." should be changed to read "... we find that the relative effectiveness of each..."*

The text has been modified appropriately.

5. *On page 5, line 5: "From inspection of the human and SANN generated solutions shown in Fig. 2a,c,..." should be changed to read "From inspection of the human and SANN generated solutions shown in Figs. 2a and c respectively,..."*

Text has been modified appropriately.

6. *It is clear from Fig. 2c that the current optimum sequence swings between the boundary values. The paper would be improved if the authors could pass comment on whether increasing the span of these boundary values would potentially yield clouds with a higher OD.*

Currently we are limited by the bounds of the experimental setup and hope in the future to test this premise with an improved experimental design. We have modified the manuscript by adding additional comments regarding the boundaries to the end of paragraph 2 pg 5.

7. Figure 2 would be improved if the y axis labels in Figs. 2d and e were given different labels rather than simply ‘transmission’. This would stress that the transmission values in these plots were measured using different detectors, transitions and detunings.

The axis labels in Figs 2d and e have been modified to read ‘Axial Transmission’ for the on axis probe measurement, ‘Transverse Transmission’ for the side imaging and ‘Integrated Transmission’ for the integrated cross-section. A clarification has also been made in the figure caption.

Response to Referee 2

Major Concern:

The major concern I have is understanding/justifying the irregular compression sequence found by the neural network. The authors state (or at least, strongly imply) that this sequence is the result of accounting for complex dynamics in the MOT. However, it is not clear to me that this is not simply a result of overfitting. Given that the network is dense and deep, there should be a very large number of variables/weights, which could overfit to the limited amount of training data (effectively allowing the network to individually account for each training data set, rather than extracting general optimization trends/processes). It seems that doing so would result in exactly the kind of irregular compression sequence that is observed, as well as many other local minima with nearly the same effectiveness (also observed). The observation that the optimal values at many of the sequence times lie on the boundary of the allowed values could also point to overfitting a limited data set. The fact that the early stopping technique had to be used is concerning for the same reason. Of course, there is no doubt that the network produced an improved solution – but whether that solution is accounting for complex atomic physics dynamics or just experimental variations in the training data (or other experimental aspects/imperfections difficult to measure directly) is not at all clear.

It is our opinion that the results themselves strongly imply complex dynamics in the MOT, however we refrain from significant speculation on that point. Instead, the present work focuses only on reporting that the SANN outperformed our best efforts as experimentalists to optimise the MOT manually. Our efforts were based both on a simple understanding of MOT dynamics as well as on experiential intuition gained from working with the system over the past four years. Neither the SANN optimisation or our manual efforts represent an exhaustive study of the MOT dynamics and we do not wish to present our findings as such here. Rather, we believe that the solutions found by the SANN indicate a potentially interesting avenue of research into the MOT dynamics.

Out of interest, we have allowed the SANN to optimise the MOT when restricted to only using adiabatic ramps, as well as restricting the number of points available. In these cases, the SANN found good solutions, but none that showed as high an optical depth as those presented in the manuscript. This seems to suggest that the complex solutions have a real advantage over smooth ramps, however our data is not yet complete. We intend to perform a more comprehensive study of the MOT compression dynamics, but chose not to include preliminary results in this work for the sake of clarity.

We have included a paragraph in the methods section discussing the fact that the found solutions are stable over a number of months and seem robust to small variations in experimental conditions. Indeed, the best solution was found in our second optimisation and it has consistently performed better than any other compression sequence even after re-aligning portions of the trapping lasers months later. This compression sequence was also used in a number of memory experiments, allowing us to compare memory efficiency between compression sequences and providing an independent measure of the relative optical depths produced by different sequences. The results were again consistent that the SANN solution was the best that we have used.

We have added an additional section to the methods titled Network Behaviour and Convergence. The section provides greater detail on how the algorithm uses overfitting to explore the cost landscape and added figure 4 to show how it converges on well-behaved solutions as more data is collected.

Other concerns:

1. *Boundary effects: The fact that the optimized values for the compression sequence are often at the boundary of each control variable's range is a concern. Have the authors considered the possibility that the true optimal solution actually lies outside of this range of values?*

Currently we are limited by the bounds of the experimental setup and hope in the future to test this premise with an improved experimental design. We have modified the manuscript by adding additional comments regarding the boundaries to the end of paragraph 2 pg 5.

2. *Local minima: The authors note that there are many local minima with OD within a few percent of the "optimal" one. How similar are these other compression sequences to "optimal" solution presented? In the Methods, the authors note these had "slightly different parameter profiles" – please be more quantitative here. Presenting and discussing some of these solutions (perhaps in supplementary information) could be very useful.*

We have included the outcomes from four other optimisations (we have only performed five complete optimisation runs due to the time taken by each) in the supplementary material. So far, we have no conclusions that we were able to draw from the the different compression sequences, but we hope to explore this much more exhaustively in future work.

3. *Human optimization: On page 2, column 2, 13 lines from the bottom, the authors note the "human optimized" solutions are linear ramps with the end points empirically optimized. Were other smoothly varying ramps (e.g. tanh, exp, gaussian) attempted? (Limiting the "human optimized" parameter profiles to linear ramps seems to limit the extent to which all adiabatic ramps can be excluded as non-optimal.)*

The optimisation of the linear ramps that we have performed for our experimental setup has consisted of a combination of empirical optimisation and automatic optimisation using a Gaussian Process approach. The automatic optimisation explored linear profiles that found not only allowed us to adjust the end points of the ramp but also the ramp time. During this optimisation we found in our case this optimal ramp was not out performed by tanh ramps which were optimised in previous memory experiments. However we do not wish to claim that this solution should rule out all adiabatic ramps as being non-optimal but prefer to say that for our case we haven't found adiabatic ramps that come even close to our solution which may hint at unexplored dynamics.

4. *Parameters: The authors refer to "63 parameters" throughout most of the manuscript. For the reader, please clarify early in the manuscript (e.g., in the first paragraph on page 2) that these are 3 control variables, with values set at 21 distinct times (which together then constitute 63 overall parameters to optimize).*

Clarified in the first paragraph of page 2 that there are 3 control variables divided into 21 time bins.

5. *pg. 2, column 1, end of second paragraph: It is unclear to me why the training time for the ANN scales linearly – please cite a reference here.*

Have modified the misleading sentence to imply that an ANN may scale linearly in certain implementations and then referenced the Goodfellow's deep learning book which contains all the theory.

6. *pg. 2, column 2, 3 lines from bottom: The authors state here that the SANN has arbitrary experimental control, when it seems the values are actually limited to a certain range. Please clarify the statement here.*

Paragraph 1 on page 3 has been modified to clarify that the bounds due to the experiments physical limitations are the only constraints imposed.

7. *pg. 4, column 1, line 8: Here the authors say the experiment repeats the compression sequence. Presumably, this is just to amplify the effect of the parameter set on the final OD, since I assume only one OD measurement is completed in each case (following the remainder of the experimental stages, such as polarization gradient cooling, etc.). Please clarify this here.*

This point was also raised by Referee 1. We have addressed this issue in paragraph 2 pg 4 by clarifying that is the entire experimental run that is repeated at 2Hz and not the compression sequence on its own. We also clarify the equilibrium condition and how it is related to atomic loss. Figure 3 has also been added to the Methods section to clarify the steps in an experimental run (including the use of PGC and optical pumping) and how the experimental run is repeated.

Response to Referee 3

1. *I did not find the flow of the manuscript in describing the steps of the proposed method that satisfying. For example authors explain how they use the Thompson sampling to choose the best model out of three ANNs that they train. However they fail to explain how they use it when it comes to the technical part of the algorithm. It also does not show-up in the Fig.1 as an important part of the control loop. This happened with the role of DE in whole procedure as well.*

We have made the description of the algorithm more explicit in the main text. We now state that we test the predictions from each of the three ANNs along with one prediction from the DE algorithm and add all four new data points to the training set. We have also updated the methods section to clarify the use of DE in the algorithm and our implementation of Thompson sampling starting at the final paragraph of pg 7. It should now contain much more explicit details regarding training and hyperparameters.

2. *From what I understand authors use the whole dataset as training set without having an development or test sets. If this is true then I suspect that the convergence of the algorithms come from ANNs because without having the test/dev sets then each ANNs will overfit to the training data and there is no guarantee that each would represent a fair approximate model of the cost function. On the other hand if authors have considered the train/test/dev set in their Machine Learning procedure then it has to be clearly explained in the manuscript.*

In response to this and similar comments from Referee 2 we have added an additional section to the methods section regarding the network behaviour and our measure of convergence. Here we specify that that we don't use the train/test/dev model as we can immediately test inferences made by the model on the physical system which are then added to the training data. Here we discuss that initially we rely on over-fitting in the model to facilitate exploration of the parameter space. As more of the space is explored however these incorrect predictions become less and less as new information is added to the training data continually after each prediction and the nets are retrained. As we are using the neural networks for optimisation purposes we do not rely on the model that the SANN contains to accurately represent the systems response across the entire cost domain. Where the parameter sets produce little atomic density tend to be undersampled by the nets as that is less interesting for our purposes. If the model is overfitting to the data then this would manifest as a tendency to sample from suboptimal areas of the space which affect the speed and ability of our model to converge, which is measured by agreement between the networks. However we can look at the predicted cross-sections of the landscape (new Figure 4) to determine the system response predicted by the SANN and we find the response to be smooth as expected from theoretical behaviour and also experimental observations from manually tuning the solutions.

3. *In order to make the work reproducible one needs to know the set of hyperparameter used in this work. This includes the parameter for DE (mutation, crossover and so on) or the percentage of train/test/dev sets, learning rates and schedule if there is any and etc.*

As noted for the first comment we have added additional information to the methods section regarding the missing hyperparameters that will make the model reproducible.

4. *Fair comparison. One way authors can benchmark their method is to compare it with other quantum control procedures like, Krotov, GRAPE, DE, etc. Comparing the quality and speed of these algorithms*

with the proposed DeepLearning approach will gives us a better measure to judge the performance of the proposed method.

This is something that we intend to do in the future however at the current time benchmarking this algorithm against current algorithms would require us to undertake a number of months of experimental work due to the complicated nature of the experiment. With an improved experimental design we hope to improved the stability and duty cycle which will allow us to perform these bench marking results. We feel we are careful not to make any claims regarding how fast/efficient the algorithm is against other algorithms other than gaussian process which we have used previously. We hope to highlight the interesting implications that arise from the solutions that seem to defy physical intuition.

REVIEWERS' COMMENTS:

Reviewer #1 (Remarks to the Author):

The manuscript by Tranter and colleagues has been significantly revised to accommodate suggestions made by the referees. The experimental sequence is now very clearly presented to the reader and the description of the algorithm is more explicit. Overall, I am very satisfied with the changes made to this article and recommend its publication.

Reviewer #2 (Remarks to the Author):

I thank the authors for clarifying the points raised in my previous report, and especially appreciate the section added to the Methods (and the added figures in the Methods). I believe the results are now clearly presented, and allows the reader to render an informed conclusion. In addition, I believe the results are significant enough to warrant publication in Nature Communications, and I now recommend publication.

Suggestion: Since the authors do not claim to rule-out all adiabatic ramps as non-optimal, the final suggestion I have for the authors is to revise the statement in paragraph 1 on page 1 (lines 18-24) which currently reads "The solution identified by machine learning is radically different to the smoothly varying adiabatic solutions currently used. Despite this, the solutions vastly outperform best known solutions producing higher optical densities." I suggest being very slightly more specific about the solutions compared here. Perhaps the authors could consider the modified statement "The solution identified by machine learning is radically different to the smoothly varying adiabatic solution commonly used. Despite this, the solution vastly outperforms the conventional adiabatic solution producing higher optical densities.", or some similar revision. This phrasing makes clear there is one style of adiabatic ramp compared to, and seems more in-line with the statement at the end of page 2, where the authors describe the human-optimized solution.

Reviewer #3 (Remarks to the Author):

Given the current status of the manuscript I do believe that the current work meets all the requirements for publication in Nature Communication Journal. Authors have kindly given enough information with respect the reproducibility concerns that I had. The flow of the draft is much more clear and algorithms are explained in a very systematic fashion. I strongly suggest to consider the current work for publication.